# Intestinal Permeability Study of Clinically Relevant Formulations of Silibinin in Caco-2 Cell Monolayers

**DOI:** 10.3390/ijms20071606

**Published:** 2019-03-31

**Authors:** Almudena Pérez-Sánchez, Elisabet Cuyàs, Verónica Ruiz-Torres, Luz Agulló-Chazarra, Sara Verdura, Isabel González-Álvarez, Marival Bermejo, Jorge Joven, Vicente Micol, Joaquim Bosch-Barrera, Javier A. Menendez

**Affiliations:** 1Instituto de Biología Molecular y Celular (IBMC) and Instituto de Investigación, Desarrollo e Innovación en Biotecnología Sanitaria de Elche (IDiBE), Universidad Miguel Hernández (UMH), 03202 Elche, Spain; almudena.perez@umh.es (A.P.-S.); vruiz@umh.es (V.R.-T.); lagullo@umh.es (L.A.-C.); 2Program Against Cancer Therapeutic Resistance (ProCURE), Metabolism and Cancer Group, Catalan Institute of Oncology, 17007 Girona, Spain; ecuyas@idibgi.org (E.C.); sverdura@idibgi.org (S.V.); 3Girona Biomedical Research Institute (IDIBGI), 17190 Girona, Spain; 4Pharmacokinetics and Pharmaceutical Technology Area, Engineering Department, Universidad Miguel Hernández (UMH), San Juan de Alicante, 03202 Alicante, Spain; isabel.gonzalez@umh.es (I.G.-Á.); mbermejo@umh.es (M.B.); 5Unitat de Recerca Biomèdica, Hospital Universitari Sant Joan, Institut d’Investigació Sanitària Pere Virgili, Universitat Rovira i Virgili, 43201 Reus, Spain; jjoven@grupsagessa.com; 6CIBER, Fisiopatología de la Obesidad y la Nutrición, CIBERobn, Instituto de Salud Carlos III (CB12/03/30038), 07122 Palma de Mallorca, Spain; 7Department of Medical Sciences, Medical School University of Girona, 17003 Girona, Spain; 8Medical Oncology, Catalan Institute of Oncology (ICO), Dr. Josep Trueta University Hospital, 17007 Girona, Spain

**Keywords:** silibinin, cancer, bioavailability, blood–brain barrier

## Abstract

An ever-growing number of preclinical studies have investigated the tumoricidal activity of the milk thistle flavonolignan silibinin. The clinical value of silibinin as a bona fide anti-cancer therapy, however, remains uncertain with respect to its bioavailability and blood–brain barrier (BBB) permeability. To shed some light on the absorption and bioavailability of silibinin, we utilized the Caco-2 cell monolayer model of human intestinal absorption to evaluate the permeation properties of three different formulations of silibinin: silibinin-meglumine, a water-soluble form of silibinin complexed with the amino-sugar meglumine; silibinin-phosphatidylcholine, the phytolipid delivery system Siliphos; and Eurosil^85^/Euromed, a milk thistle extract that is the active component of the nutraceutical Legasil with enhanced bioavailability. Our approach predicted differential mechanisms of transport and blood–brain barrier permeabilities between the silibinin formulations tested. Our assessment might provide valuable information about an idoneous silibinin formulation capable of reaching target cancer tissues and accounting for the observed clinical effects of silibinin, including a recently reported meaningful central nervous system activity against brain metastases.

## 1. Introduction

Silymarin is an extract of *Silybum marianum* (milk thistle) seeds that was classified by the World Health Organization in the 1970s as an official medicine with health-promoting properties. It is a mixture of several flavonoids (e.g., taxifolin, quercetin, kaempferol, and apigenin) and seven flavonolignans, including silychristin A, silychristin B, silydianin, silybin B, silybin A, isosilybin A, and isosilybin B [1,2,3,4,5,6,7]. The highest concentration of silymarin (50–70% in the extract and 20–40% in commonly used pharmaceutical preparations) corresponds to silybin or silibinin (CAS No. 22888-70-6, a 1:1 mixture of the diasteroisomers silybin A and silybin B), which is considered as the major bioactive component of silymarin [3,8,9]. 

Despite containing several hydrophilic ionizable groups, the overall character of silibinin is hydrophobic, with very poor solubility in water. Silibinin is soluble in polar aprotic solvents (e.g., acetone, N,N-dimethylformanide [DMF], and tetrahydrofuran [THF]), but is poorly soluble in polar protic solvents (e.g., ethanol and methanol), and is insoluble in non-polar solvents (e.g., chloroform and petroleum ether) [8]. With respect to pharmacokinetics, silibinin is quickly absorbed after oral administration and exhibits a good distribution in a variety of tissues, including liver, lung, stomach, skin, and small bowel. However, the half-life of silibinin elimination via excretion in bile and urine is very fast and oscillates between one and three hours [10]. Moreover, it has been suggested that silibinin is too large to be absorbed by simple diffusion and has poor miscibility with other lipids, thereby reducing its capacity to cross the lipid-rich outer membrane of the enterocytes of the small intestine. In addition, silibinin undergoes extensive phase II metabolism through the first liver passage after its absorption [11]. These four factors (low water solubility, rapid excretion, inefficient intestinal absorption, and elevated metabolism) significantly decrease the hematic concentration of silibinin when combined, reducing its arrival at the target organ and, consequently, limiting its therapeutic efficiency [12,13,14]. 

Numerous approaches have been pursued to overcome the low bioavailability of silibinin through the development of an enormous number (>200) of silibinin modifications [8,15,16,17]. These modifications can be grouped into four main categories, namely: complexation with -cyclodextrins, chemical modification to generate different derivatives with enhanced water-solubility profiles (e.g., phosphate and sulphate salts, glycol-conjugates [gluco-, manno-, galacto-, and lacto-conjugates]), incorporation into different delivery technologies (e.g., solid dispersions, floating tablets, softgel capsules, micronizated/nanonizated formulations, etc.), and self-microemulsifying drug delivery systems (e.g., microspheres, nanoparticles, micelles, and phytosomes). The increased absorption of some of these new formulations has confirmed the high tolerability of silibinin, with no deaths or life-threatening adverse events reported during its therapeutic use [13,18]. 

To shed some light on the absorption and bioavailability capabilities of different silibinin formulations with proven pre-clinical and clinical activity, we evaluated the permeation properties of three different silibinin formulations: silibinin-meglumine, a more water-soluble form of silibinin complexed with the excipient amino-sugar meglumine [19,20]; silibinin-phytosome (Siliphos), a silibinin–phosphatidylcholine (PC) complex that can be administered to humans at doses achieving micromolar concentrations with minimal or no side effects [21,22,23]; and Eurosil^85^/Euromed, a patented extract of milk thistle ETHIS-094 that is the active component of the nutraceutical Legasil with enhanced bioavailability [24,25]. Intestinal permeability was evaluated using the Caco-2 cell monolayer model, a well-accepted model of human intestinal absorption [26,27,28,29,30].

## 2. Results

We first established the Caco-2 cell monolayer on a permeable transwell filter support and tested its integrity and reliability by measuring the transepithelial electrical resistance (TEER) as a function of time, which remained stable during the assays irrespective of the silibinin formulation assayed (Appendix A). We then added the silibinin formulations and sampled the apical (AP) and basolateral (BL) compartments at different time points in the assay. None of the concentrations of the different silibinins employed were toxic to the cells as measured by 3-(4,5-dimethylthiazol-2-yl)-2,5-diphenyltetrazolium bromide (MTT)-based cell viability assays (Appendix A).

Samples were analyzed by high-performance liquid chromatography (HPLC)–UV absorbance to quantify the amounts of silibinin crossing the cell monolayer (Figure 1A). These concentration values were employed to obtain the apparent permeability values (*P*_app_). To validate the Caco-2 monolayer system, we measured the *P*_app_ of metroprolol, a well-transported marker by passive diffusion, across the Caco-2 monolayer from the AP to BL chamber. The *P*_app_ value of metroprolol was determined as 6 × 10^−5^ cm/s, which is expected for well-absorbed drugs [31,32]. The calculated *P*_app_ values for the different formulations of silibinin across the monolayer in both directions (AP–BL and BL–AP), as well as the efflux ratios (BL–AP/AP–BL), are shown in Figure 1B,C and Figure 2A, respectively. 

The highest permeation values for the AP–BL direction were observed for the Eurosil^85^/Euromed formulation (3.3 × 10^−6^ cm/s; Figure 1A), which was the sole formulation reaching a similar value to that observed when using a pure standard of silibinin (3.2 × 10^−6^ cm/s). Moreover, there was no indication of efflux or active transport according to the net efflux criterion proposed by the Food and Drug Administration guidelines, as the ratio of *P*_app_ BL–AP/*P*_app_ AP–BL for the Eurosil^85^/Euromed formulation was 1.57 (<2), significantly lower than that obtained with silibinin-PC (Figure 2A). 

Regarding the BL–AP direction, the silibinin-phosphatidylcholine complex Siliphos demonstrated the highest permeation value (6.29 × 10^-6^ cm/s; Figure 1B), which was similar to the value observed when using a pure standard of silibinin (6.28 × 10^−6^ cm/s). Moreover, this formulation showed an efflux ratio notably greater than 2 (3.3), therefore suggesting an active secretion transport (Figure 2A). 

Because there exists a good correlation between the Caco-2-derived *P*_active_ value (i.e., [*P*_app_ BL–AP − *P*_app_ AP–BL]/2) and in situ blood–brain barrier (BBB) permeation rate, the Caco-2 assay can be used to indirectly evaluate BBB potential for compounds with *P*_active_ < 5 × 10^−6^ cm/s. The Eurosil^85^/Euromed formulation was the only one exhibiting a *P*_active_ < 5 (3.5) (Figure 2B), thereby suggesting a higher potential to permeate the BBB compared to silibinin-PC. 

## 3. Discussion

Originally described as a remedy for the bites of poisonous snakes more than 2000 years ago, the use of silibinin-containing nutraceuticals to treat liver toxicity, including alcoholic liver disease, nonalcoholic liver disease, drug-induced liver injury, cirrhosis, viral hepatitis, and mushroom poisoning, has been well documented over the last 40 years [2,17,33]. In the last decade, numerous studies have begun to demonstrate the capacity of silibinin to exert significant tumoricidal activity against cultured cancer cells and xenografts, to enhance the efficacy of other anti-cancer therapeutic agents, to reduce the toxicity of cancer treatments, and to prevent and overcome the emergence of cancer drug resistance [34,35,36,37]. 

Despite this ever-growing number of preclinical studies showing the capacity of silibinin to target tumor cells, the achievement of a bona fide, clinically relevant anti-cancer activity of silibinin remains controversial in human trials [38]. This could be explained by the poor water solubility (<0.04 mg/mL) of its flavonolignan structure and subsequent low bioavailability. Not surprisingly, many methods have been developed to improve the solubility and bioavailability of silibinin. In this regard, the aim of the present study was to compare the intestinal absorption of silibinin in different pre-clinical and clinically relevant formulations in the Caco-2 model, which expresses intestinal efflux and uptake transporters that regulate permeation of drugs from intestinal lumen to systemic circulation [26,27,28]. 

Generally, compounds with *P*_app_ < 1 × 10^−6^ cm/s, *P*_app_ 1–10 × 10^−6^ cm/s, and *P*_app_ > 10 × 10^−6^ cm/s can be classified as poorly (0–20%), moderately (20–70%) and well- (70–100%) absorbed compounds, respectively, in the Caco-2 model [31,32]. In our hands, all the *P*_app_ values of the different silibinin formulations tested were at a level of 10^−6^ cm/s, and so they can be assigned to the moderately absorbed group of compounds. The AP–BL and BL–AP trends between silibinin formulations were significantly different when comparing silibinin-PC and Eurosil^85^/Euromed. Indeed, a correlation appeared to exist between the efflux mode and the reported anti-cancer effect of silibinin formulations in a clinical setting. In this respect, early studies suggested that the effect of enhanced bioavailability achieved with the phytolipid delivery system—a formulation that was initially named silipide (IdB 1016) or Siliphos—was likely related to the passage of the silibinin phosphatidylcholine complex through the gastrointestinal tract [3,39,40,41,42]. However, although high-dose oral silybin-phytosome has been shown to achieve transient high blood concentrations, low levels of silibinin and no significant anti-tumor activity were reported in prostate cancer tissue [19,20]. The results of our analysis predict the active secretion transport of the Siliphos formulation. On the other hand, when used as part of Legasil—a commercially available nutraceutical product containing the Eurosil^85^/Euromed formulation—silibinin has recently been shown to exhibit significant clinical activity in cancer patients with advanced systemic disease [43,44,45]. Indeed, responses to Eurosil^85^/Euromed-based therapy were notable in the central nervous system, where highly significant clinical and radiological improvements of brain metastases (including several complete responses) were achieved in patients with non-small cell—lung cancer [44,45]. According to our results, the silibinin formulation exhibiting the highest permeability rate (i.e., Eurosil^85^/Euromed) was predicted to also exhibit a passive diffusion mechanism of transport [46,47]. Because some nutritional modalities are known to impact intestinal permeability, it might be relevant to evaluate how certain dietary approaches with proposed applications in oncology (e.g., high-fat, low-carbohydrate ketogenic diets, [48] might represent a potentially promising strategy to increase biodisponibility and the efficacy of silibinin-based anti-cancer strategies. Moreover, the Caco-2 data can be used to predict BBB permeability. For compounds that are not subject to significant levels of efflux activity in Caco-2 cells, there is a clear correlation between the *P*_active_ value and the permeability–surface area product (logPS) of drugs known to permeate the BBB. Considering this correlation, our findings suggest that Eurosil^85^/Euromed, but not the phytolipid delivery system, could be considered a good candidate to cross the BBB. 

The results presented here represent a new contribution to our rudimentary knowledge of the oral absorption and bioavailability of clinically relevant formulations of the flavonolignan silibinin. Our findings might provide valuable information to help identify the best silibinin formulation that would reach the target (cancer) tissues and would account for the clinical (anti-cancer) effects of silibinin, including the meaningful central nervous system activity against brain metastases. 

## 4. Materials and Methods

### 4.1. Chemicals and Reagents

All chemicals were of analytical reagent grade and were used as received. For mobile phase preparation, trifluroacetic acid (TFA) and acetonitrile were purchased from Merck (Millipore, Darmstadt, Germany) and VWR (Barcelona, Spain), respectively. Dimethyl sulfoxide (DMSO), metoprolol and standard compound silibinin were purchased from Sigma-Aldrich (Steinheim, Germany). Silibinin/phospholipids (Siliphos) was obtained from Indena S.p.A (Milan, Italy). Monteloeder (Elche, Alicante, Spain) provided the water-soluble milk thistle extract in its silibinin-meglumine salt, and Eurosil^85^/Euromed was kindly provided by Meda Pharma S. L. (Barcelona, Spain). Hank’s Balanced Salt Solution (HBSS), Dulbecco´s Modified Eagle’s Medium (DMEM), fetal bovine serum (FBS), penicillin/streptomycin, MEM Non-Essential Amino Acids (NEAA) Solution (100×) and 1 M HEPES were obtained from Gibco/Thermo Fisher Scientific (Waltham, MA, USA). The human colon adenocarcinoma cell line Caco-2 was obtained from the American Type Culture Collection. Caco-2 cells were cultured in DMEM containing D-glucose (4.5 g/L) and supplemented with 10% FBS, 1% NEAA, 1% HEPES, penicillin (100 U/mL), and streptomycin (100 µg/mL) at 37 °C in a humidified atmosphere with 5% CO_2_. 

### 4.2. Cell Viability Assay

The cytotoxic effects of the different formulations of silibinin were tested and compared with those of standard silibinin using the 3-(4,5-dimethylthiazol-2-yl)-2,5-diphenyltetrazolium bromide (MTT) assay. Caco-2 cells were seeded in 96-well plates (Costar, Fisher Scientific, Pittsburgh, PA, USA) until cell monolayers were formed. Cells were treated with different concentrations of silibinin formulations (0–200 µg/mL) or standard silibinin (0–200 µM) for 2 h. The medium was removed, and cells were then incubated with MTT for 3–4 h at 37 °C and 5% CO_2_. Then, the medium was removed and 100 µL of DMSO per well was added to dissolve the formazan crystal. The plates were shaken for 15 min and absorbance was measured using a microplate reader (SPECTROstar Omega, BMG LabTech GmbH, Ortenberg, Germany) at 570 nm.

### 4.3. Cell Culture and Permeability Studies

Caco-2 cells were seeded at a density of 1 × 10^5^ onto Transwell 6-well inserts with a polyethylene terephthalate membrane (0.4 µm pore size; BD Falcon) and were maintained at 37 °C under 90% humidity and 5% CO_2_. The medium was replaced every two to three days for both the AP and BL chambers. Cell monolayers were used 19–21 days after seeding, once confluence and differentiation were achieved. The integrity of each cell monolayer was assessed by measuring the TEER before and after the experiments with an epithelial voltammeter (Millicell-ERS). Permeability studies were performed by adding the silibinin formulations at 200 µg/mL in HBSS/0.6% DMSO (stocks were prepared at 30 mg/mL in 100% DMSO) and, in parallel, standard silibinin at different concentrations (10, 20, 50, 100, 150, 200, and 300 µmol/L). 

The transport experiment was initiated by removing the culture medium from the AP and BL chambers. The Caco-2 monolayers were washed twice with pre-warmed HBSS (pH 7.4) and incubated with the same solution at 37 °C for 30 min. The test compounds were added to the AP (2.2 mL) or BL (3.2 mL) chambers, while the receiving chamber contained the corresponding volume of HBSS. The six-well plate containing the cell monolayers was placed in an orbital environmental shaker, which was maintained at a constant temperature (37 °C) and agitation rate (54 rpm) for the duration of the transport experiments. 

To follow transport across the cell monolayer, several samples of 200 µL were collected at different time points (0, 30, 60, 90, and 120 min) from the AP or BL chambers during the permeability assay. The volume of the samples taken at each time point was replaced with the same volume of HBSS to maintain the total volume in the chamber throughout the experiment. Also, two samples of 200 µL were taken from the donor chamber, at the beginning and the end of the assay, for the mass balance calculation. 

Transport studies were performed from apical-to-basolateral (AP–BL) and basolateral-to-apical (BL–AP) chambers. The apparent permeability (*P*_app_) values for each compound were calculated according to the following equation: $$P_{\mathrm{app}}=\frac{dQ}{dt}\cdot \frac{1}{A\cdot C_{0}\cdot 60}$$
where *P*_app_ is the apparent permeability (cm/s), *dQ*/*dt* is the steady-state flux, *A* is the diffusion area of the monolayers (cm^2^), *C*_0_ is the initial concentration of the drug in the donor compartment (µM), and 60 is a conversion factor [49]. 

The efflux ratio was calculated to determine the absorption mechanism such as the ratio of *P*_app_ (BL–AP) − *P*_app_ (AP–BL).

### 4.4. Analytical Methodology

Analyses were performed using an Agilent LC 1100 series HPLC system (Agilent Technologies, Inc., Palo Alto, CA) controlled by Chemstation software, equipped with a pump, autosampler, column oven, and UV–VIS diode array detector (wavelength selected at 280 nm to detect silibinin). The samples were separated on a Poroshell 120 SB-C18 column (2.7 µm, 4.6 × 150 mm). The flow rate was 0.5 mL/min, the column temperature set at 22 °C, and the mobile phases consisted of 0.1% TFA in water as mobile phase A and acetonitrile as mobile phase B, using a gradient elution based on the following profile: 0 min, 25% B; 5 min, 40% B; 10 min, 50% B; 15 min, 25% B; 20 min, 25% B. Quantitation of the silibinin concentration was performed using a commercial standard. A calibration graph for the quantitative evaluation of silibinin was performed using a six-point regression curve (r^2^ > 0.999).

### 4.5. Statistical Analysis

One-way analysis of variance and statistical comparisons of the different treatments were performed using Tukey´s post-test in GraphPad Prism version 6.00 (GraphPad Software, San Diego, CA, USA).

## Figures and Tables

**Figure 1 ijms-20-01606-f001:**
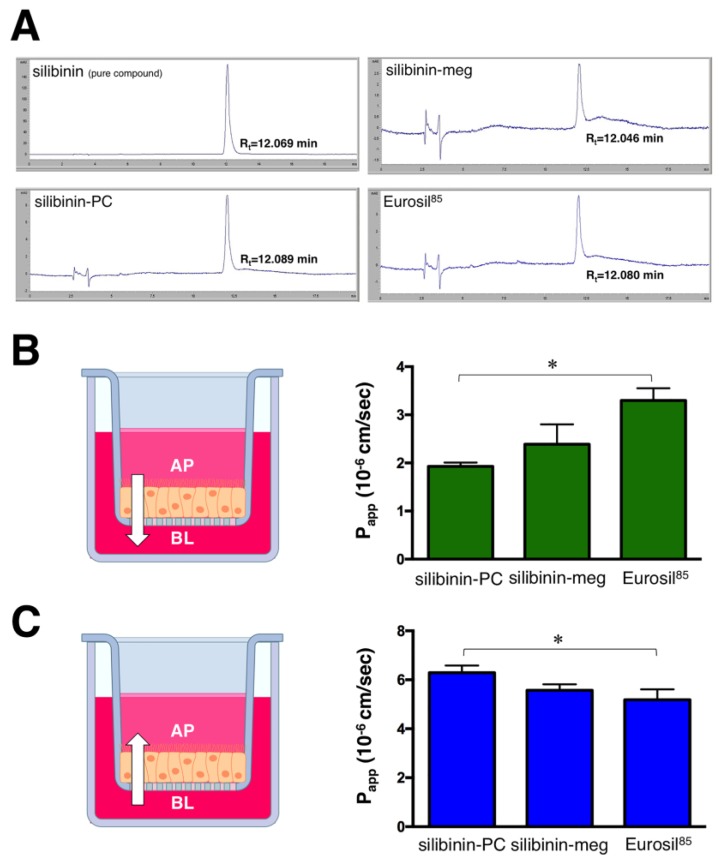
*P*_app_ values of the different formulations of silibinin. Representative HPLC elution profiles and retention times of silibinin formulations after 120 min incubation with the Caco-2 cell monolayers (**A**). *P*_app_ values in cm/s for silibinin formulations in both AP–BL (**B**) and BL–AP (**C**) directions. Each column represents the mean ± standard deviation (SD) of *P*_app_ values obtained in n = 6 independent replicates. * One-way ANOVA *p* < 0.0001; AP: Apical; BL: Basolateral; silibinin-PC: silibinin-phosphatidylcholine; silibinin-meg: silibinin-meglumine).

**Figure 2 ijms-20-01606-f002:**
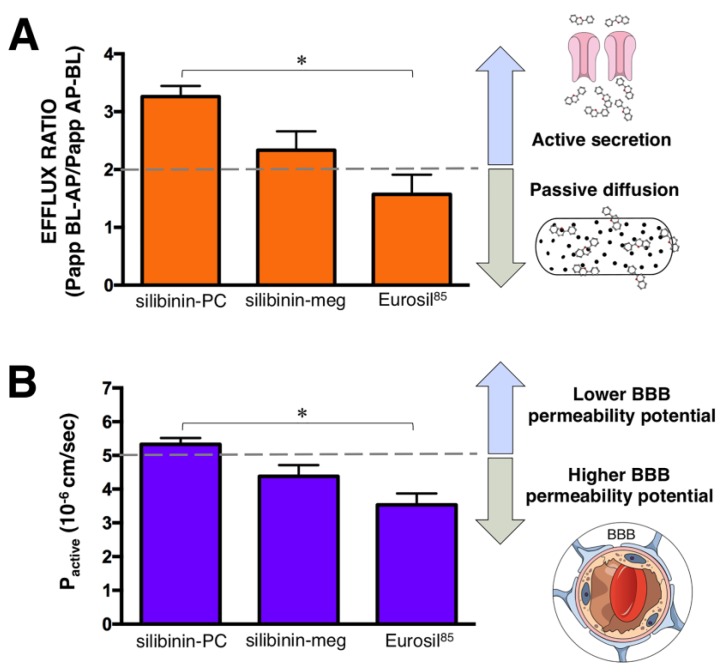
Transport ratios of the different formulations of silibinin. (**A**) Efflux ratio (*P*_app_) BL–AP/*P*_app_ AP–BL). (**B**) Blood–brain barrier (BBB) permeability-related *P*_active_ ratio ([*P*_app_ BL–AP − *P*_app_ AP–BL]/2). * One-way ANOVA *p* < 0.0001; silibinin-PC: silibinin-phosphatidylcholine; silibinin-meg: silibinin-meglumine.

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
