# Peer review of "Intestinal Permeability Study of Clinically Relevant Formulations of Silibinin in Caco-2 Cell Monolayers"

_ijms, 2019, doi:10.3390/ijms20071606_

Reviewer 1 Report

JournalIJMS (ISSN 1422-0067)

Manuscript ID ijms-474428

Type Brief Report

Number of Pages 14

Title Intestinal permeability study of clinically relevant formulations of silibinin in Caco-2 cell monolayers

Authors Almudena Pérez-Sánchez , Elisabet Cuyàs , Verónica Ruiz-Torres , Luz Agulló-Chazarra , Sara Verdura , Isabel Álvarez-González , Maria del Val Bermejo , Jorge Joven , Vicente Micol * , Joaquim Bosch-Barrera * , Javier A. Menendez *

Comments to the Author:

The manuscript describes differences in the transport and blood-brain barrier permeabilities between 3 silibinin formulations using an in vitro model of human enterocytes.

Overall, the manuscript is well written with adequate description of the methods/procedures and presentation of data in an organized manner. However, some concerns exist, which should be addressed in a revised manuscript as indicated below.      

Minor concerns

1) Figure 1 B: The order of the 3 formulations is changed in this figure. Please be coherent along the manuscript. Change “P acrive” for “P active” in Y axis of the same figure.

2) Line 139: cm/s or cm/sec, please maintain the style along the manuscript.

Major concerns

1) In any of the figures the error bars are shown. Although the authors indicated that there are no statistically significant differences the error bars would give more information.

2) Lines 161-162: In line with the last concern, why do you consider a better candidate one or another formulation if there are not significant differences between them?

3) Lines 211-213: The authors followed the transport across the cell monolayer at different time points (0, 30, 60, 90 and 120 min) from the AP or BL chambers during the permeability assay, however these results are not shown. These time course data would be of great interest to see if the absorption is time dependent or the system is saturated.

4) Is the concentration reached basolaterally clinically relevant? Does it reach the effective doses reached in patients?

4) How did the authors chose the concentrations tested?

5) There are food/drug interactions that might enhance drug transport. For example, high lipid based foods might increase the transport of some drugs. Do you think this might be applied also to sibilinin formulations?

Author Response

We have now carefully followed all the suggestions offered by the anonymous reviewers. Below, we have addressed all the comments and questions raised by reviewers point-by-point. We have made changes in the text accordingly (in red). The original reviewers’ comments are italicized and our responses and actions to these comments follow (in bold). 

Reviewer #1

Comments to the Author:

The manuscript describes differences in the transport and blood-brain barrier permeabilities between 3 silibinin formulations using an in vitro model of human enterocytes.

Overall, the manuscript is well written with adequate description of the methods/procedures and presentation of data in an organized manner. However, some concerns exist, which should be addressed in a revised manuscript as indicated below.      

We thank the reviewer for the positive evaluation of our manuscript.

Minor concerns

1) Figure 1 B: The order of the 3 formulations is changed in this figure. Please be coherent along the manuscript. Change “P acrive” for “P active” in Y-axis of the same figure.

Thank you for noticing it. The order of the 3 formulations is now the same in all the multi-panel figures throughout the manuscript. 

“P acrive” has been changed to “P active” in Y-axis in Fig. 2B. 

2) Line 139: cm/s or cm/sec, please maintain the style along the manuscript.

cm/sec is now being used both in the text and the figures. 

Major concerns

1) In any of the figures the error bars are shown. Although the authors indicated that there are no statistically significant differences the error bars would give more information.

Statistically tests were mistakenly performed in the previous report as they inadvertently included data from other unrelated formulations. We sincerely apologize for such miscalculation. 

Errors bars are now included in all the multi-panel figures. 

We now report one-way ANOVA and multiple comparison tests of the 3 formulations to reveal that statistically significant differences exist between the Eurosil85/Euromed and the silibinin-PC/Siliphos formulations. 

2) Lines 161-162: In line with the last concern, why do you consider a better candidate one or another formulation if there are not significant differences between them?

Based on the current statistical analyses, the Eurosil/Euromed formulation might be viewed as a better candidate to be tested in a clinical setting. 

3) Lines 211-213: The authors followed the transport across the cell monolayer at different time points (0, 30, 60, 90 and 120 min) from the AP or BL chambers during the permeability assay, however these results are not shown. These time course data would be of great interest to see if the absorption is time dependent or the system is saturated.

The Papp values are actually computed from samples taken at different time points using time-dependent linear regression calculations. 

4) Is the concentration reached basolaterally clinically relevant? Does it reach the effective doses reached in patients?

Silibinin concentrations reached basolaterally were in the low micromolar range, which are consistent with those obtained in a clinical setting via oral administration. 

4) How did the authors choose the concentrations tested?

Concentrations were based on previous TEER and MTT-based cell viability assay using a wide range of concentrations that failed to promote significant effects on the integrity and viability of the Caco-2 monolayer (see Supplementary Fig. S1). 

5) There are food/drug interactions that might enhance drug transport. For example, high lipid based foods might increase the transport of some drugs. Do you think this might be applied also to silibinin formulations?

We now briefly acknowledge such suggestion in the Discussion section of the revised manuscript (lines 174-179): 

According to our results, the silibinin formulation exhibiting the highest permeability rate (i.e., Eurosil85/Euromed)was predicted alsoto exhibit a passive diffusion mechanism of transport [46,47]. Because some nutritional modalities are known to impact intestinal permeability, it might relevant to evaluate how certain dietary approaches with proposed applications in oncology (e.g., high-fat, low-carbohydrate ketogenic diets, 48] might represent a potentially promising strategy to increase biodisponibility and efficacy of silibinin-based anti-cancer strategies. 

Reviewer 2 Report

Authors referred to "blood-brain barrier permeabilities" But they used Caco-2 cell Line,  a colon cancer cell Line.  How the authors analyzed the   Blood-brain barrier permeability? Please verify the discrepancy.

The authors performed a statistical analysis but the differences seem not to be  significative. There is not a P-value.

 In the matherial and methods section it has been described the Cell viability assay but it is not reported in the manuscript. Please Remove it.

Did the authors used a positive and a negative control?   How they can demonstrate that the only substance present in the samples analyzed is silibinin?
  Please add the graph with the retention time.
Is it possible use an additional method to demonstrate the presence of silibinin?

Author Response

POINT-BY-POINT REBUTTAL LETTER

Ms. Ref. No.:  ijms-474428

Title: Intestinal permeability study of clinically relevant formulations of silibinin in Caco-2 cell monolayers
International Journal of Molecular Sciences

We have now carefully followed all the suggestions offered by the anonymous reviewers. Below, we have addressed all the comments and questions raised by reviewers point-by-point. We have made changes in the text accordingly (in red). The original reviewers’ comments are italicized and our responses and actions to these comments follow (in bold). 

Reviewer #2

Authors referred to "blood-brain barrier permeabilities" But they used Caco-2 cell Line,  a colon cancer cell Line.  How the authors analyzed the   Blood-brain barrier permeability? Please verify the discrepancy.

There exists a good correlation between Caco-2 derivedPactive value(i.e., [PappBL-AP - Papp AP-BL]/2) andin situblood brain barrier (BBB). 

https://www.cyprotex.com/admepk/in-vitro-permeability/caco-2-permeability

Therefore, the Caco-2 assay can be used to indirectly evaluate BBB potential for compounds with Pactive < 5 x 10-6cm/sec.

The authors performed a statistical analysis but the differences seem not to be significative. There is not a P-value.

Statistically tests were mistakenly performed in the previous report as they inadvertently included data from other unrelated formulations. We sincerely apologize for such miscalculation. 

Errors bars are now included in all the multi-panel figures. 

We now report one-way ANOVA and multiple comparison tests of the 3 formulations to reveal that statistically significant differences exist between the Eurosil85/Euromed and the silibinin-PC/Siliphos formulations. 

In the material and methods section it has been described the Cell viability assay but it is not reported in the manuscript. Please remove it.

We now provide de MTT-based cell viability assays in the Supplementary Fig. S1B. 

Did the authors use a positive and a negative control?   How they can demonstrate that the only substance present in the samples analyzed is silibinin?
Please add the graph with the retention time. 
Is it possible use an additional method to demonstrate the presence of silibinin?

Figure 1A shows now representative HPLC elution profiles and retention times of each silibinin formulation after 120 minutes incubation with the Caco-2 cell monolayers. Pure standardized silibinin was employed as positive control (DMSO v/vwas employed as negative control in all cases) to demonstrate that the sole substance present in the samples analyzed was silibinin. 

Round  2

Reviewer 2 Report

I accept the manuscript in the present manner